# Key necroptotic proteins are required for Smac mimetic-mediated sensitization of cholangiocarcinoma cells to TNF-α and chemotherapeutic gemcitabine-induced necroptosis

Perawatt Akara-amornthum[1], Thanpisit Lomphithak[1], Swati Choksi[2], Rutaiwan Tohtong[3], Siriporn Jitkaew[4]*

1 Graduate Program in Clinical Biochemistry and Molecular Medicine, Department of Clinical Chemistry, Faculty of Allied Health Sciences, Chulalongkorn University, Bangkok, Thailand, 2 Laboratory of Immune Cell Biology, Center for Cancer Research, National Cancer Institute, National Institutes of Health, Convent Drive, Bethesda, MD, United States of America, 3 Department of Biochemistry, Faculty of Science, Mahidol University, Bangkok, Thailand, 4 Age-Related Inflammation and Degeneration Research Unit, Department of Clinical Chemistry, Faculty of Allied Health Sciences, Chulalongkorn University, Bangkok, Thailand

* Siriporn.ji@chula.ac.th

## Abstract

Cholangiocarcinoma (CCA), a malignant tumor originating in the biliary tract, is well known to be associated with adverse clinical outcomes and high mortality rates due to the lack of effective therapy. Evasion of apoptosis is considered a key contributor to therapeutic success and chemotherapy resistance in CCA, highlighting the need for novel therapeutic strategies. In this study, we demonstrated that the induction of necroptosis, a novel regulated form of necrosis, could potentially serve as a novel therapeutic approach for CCA patients. The RNA sequencing data in The Cancer Genome Atlas (TCGA) database were analyzed and revealed that both receptor-interacting protein kinase 3 (RIPK3) and mixed lineage kinase domain-like (MLKL), two essential mediators of necroptosis, were upregulated in CCA tissues when compared with the levels in normal bile ducts. We demonstrated in a panel of CCA cell lines that RIPK3 was differentially expressed in CCA cell lines, while MLKL was more highly expressed in CCA cell lines than in nontumor cholangiocytes. We therefore showed that treatment with both tumor necrosis factor-α (TNF-α) and Smac mimetic, an inhibitor of apoptosis protein (IAP) antagonist, induced RIPK1/RIPK3/MLKL-dependent necroptosis in CCA cells when caspases were blocked. The necroptotic induction in a panel of CCA cells was correlated with RIPK3 expression. Intriguingly, we demonstrated that Smac mimetic sensitized CCA cells to a low dose of standard chemotherapy, gemcitabine, and induced necroptosis in an RIPK1/RIPK3/MLKL-dependent manner upon caspase inhibition but not in nontumor cholangiocytes. We further demonstrated that Smac mimetic and gemcitabine synergistically induced an increase in TNF-α mRNA levels and that Smac mimetic reversed gemcitabine-induced cell cycle arrest, leading to cell killing. Collectively, our present study demonstrated that TNF-α and gemcitabine induced RIPK1/RIPK3/MLKL-dependent necroptosis upon IAP depletion and caspase inhibition; therefore,

**Data Availability Statement:** All relevant data are within the manuscript and its Supporting Information files.

**Funding:** This work was supported by grants from National Research University Project, Office of Higher Education Commission [NRU59-030-HR] and Korea Foundation for Advanced Studies [009/2560] to SJ and the Thailand Research Fund and the Medical Research Council (UK), Newton Fund [DBG5980006] and [MR/N01247X/1] to RT. PA gratefully acknowledged the Scholarship from the Graduate School, Chulalongkorn University to commemorate the 72nd anniversary of his Majesty King Bhumibol Aduladej [GCUGE12-2] and the Chulalongkorn University 90th Anniversary Fund (Ratchadaphiseksomphot Endowment Fund) [GCUGR1125612059M]. The funders had no role in study design, data collection and analysis, decision to publish, or preparation of the manuscript.

**Competing interests:** The authors have declared that no competing interests exist.

our findings have pivotal implications for designing a novel necroptosis-based therapeutic strategy for CCA patients.

## Introduction

Cholangiocarcinoma (CCA) is an aggressive but also markedly heterogeneous malignancy originating from both intra- and extrahepatic bile duct epithelium and is also well known to be more prevalent in Asian countries, but its incidence rate has progressively increased world-wide [1, 2]. CCA generally harbors an aggressive clinical course resulting in high mortality and recurrence/metastasis rates and subsequently adverse clinical outcomes, with a relatively low 5-year survival rate (5–10%) [1, 2]. CCA is also diagnosed only when the disease has progressed to the relatively advanced clinical or pathologic stages, in which the only therapeutic option is chemotherapy without the possibility of curative surgery. Gemcitabine or gemcitabine plus cisplatin is currently the established first line of chemotherapy for CCA patients, yet the overall survival rate is still dismal, with less than one year for metastatic advanced-stage patients [3–5]. There are limited data on the molecular mechanisms of chemoresistance in this particular disease, although dysregulated apoptosis pathways have been proposed as key contributing factors [6, 7]. Therefore, the development of novel treatment strategies with targeted therapeutic approaches and the search for novel compounds that could sensitize the effects of chemotherapy are urgently required to improve the survival of CCA patients.

In addition to apoptosis, necroptosis has been recently identified as a novel form of regulated cell death elicited by various stimuli, including ligations of death ligands, pattern-recognition receptors, and chemotherapeutic agents [8]. Tumor necrosis factor-α (TNF-α) represents a well-studied *in vitro* model of necroptosis. Upon binding to TNF receptor 1 (TNFR1), TNF-α triggers multiple signaling pathways depending on the cellular context. In most cell types, TNF-α initially triggers a signaling cascade that leads to the formation of complex I, consisting of TNFR1, TRADD (TNFR1-associated death domain), TRAF2 (TNF receptor-associated factor-2), RIPK1 (receptor-interacting protein kinase 1), and cIAP1/2 (cellular inhibitor of apoptosis proteins 1/2) [9]. Activation of complex I triggers nuclear factor kappaB (NF-κB) and mitogen-activated protein kinase (MAPK), which lead to the induction of anti-poptotic proteins [such as cellular FLICE (FADD-like IL-1β-converting enzyme)-inhibitory protein or cFLIP], thereby promoting cell survival and the secretion of pro-inflammatory cytokines (such as TNF-α and IL-6) and activating inflammation. When the survival pathway is inhibited, such as via depletion of cIAP1/2, TNF-α induces the formation of complex IIa, consisting of TRADD, FADD (Fas-associated death domain-containing protein), RIPK1, and pro-caspase-8, which leads to the activation of caspase-8 and apoptosis [10]. When both cIAP1/2 and caspases are inhibited, complex IIb, also called the necrosome, is formed. The necrosome is a signaling complex composed of RIPK1 and RIPK3 (receptor-interacting protein kinase 3) as its core components [11–14]. RIPK3 then phosphorylates the pseudokinase MLKL (mixed lineage kinase domain-like), causing its oligomerization and translocation to the plasma membrane, where it disrupts the integrity of the plasma membrane [15–17]. As a consequence, necroptotic cells release their intracellular contents, also known as damage-associated molecular patterns (DAMPs) and cytokines/chemokines that render immunogenicity and activate antitumor immunity [18–20]. The results of several studies in both *in vitro* and *in vivo* models have revealed that necroptosis represents a novel target for efficient cancer therapies and provides opportunities to circumvent apoptosis resistance [18, 21]. Therefore, necroptosis appears

to be a promising novel concept for immunogenic cancer therapy, yet there are few studies on necroptosis signaling and its potential therapeutic applications in CCA. However, it is also true that many cancers subsequently develop necroptosis resistance. Acquired or intrinsic defects in necroptosis signaling pathways, as in the case of loss or reduced RIPK3 and MLKL expression in multiple cancers, are considered a major hindrance for therapeutic success at this juncture [13, 22–29].

The inhibitors of apoptosis proteins (IAPs) are key apoptosis regulators that harbor baculo-virus IAP repeat (BIR) domains [30]. In addition, a really interesting new gene (RING) domain E3 ubiquitin ligase is present in some IAPs [30]. It has been shown that second mito-chondria-derived activator of caspases (SMAC) is released from mitochondria to competitively interact with IAPs, including X-linked inhibitor of apoptosis proteins (XIAP), cIAP1, and cIAP2, through BIR domains, thereby releasing the inhibition of caspases. Smac mimetics were initially developed as small molecules mimicking the IAP-binding motif of SMAC to antagonize XIAP, thereby allowing caspase activation [31]. In addition, Smac mimetics were reported to induce the autoubiquitination and proteasomal degradation of the E3 ligases cIAP1 and cIAP2, which promote deubiquitination and release of RIPK1 from the TNFR1 complex, leading to a switch from prosurvival to death signaling complexes [32]. The depletion of cIAP1 and cIAP2 also facilitates the stabilization of nuclear factor kappaB (NF-kB)-inducing kinase (NIK) and activation of noncanonical NF-κB signaling, resulting in autocrine secretion of TNF-α [33, 34]. Overexpression of IAPs has been commonly reported in many human malignancies and frequently contributes to drug resistance by promoting evasion of cell death [35]. Preclinical studies have also demonstrated that Smac mimetics can trigger cancer cell death as single agents or as sensitizers with other anticancer drugs, including chemotherapeutic agents [30]. In addition, the combination of a Smac mimetic and TNF-related apoptosis-inducing ligand (TRAIL) has been reported to decrease cell proliferation, invasion and metastasis but did not sensitize CCA cells to cell death [36, 37].

As several Smac mimetics have currently entered clinical trials, the study of Smac mimetic-based combinatory approaches that activate necroptosis has come to be very important in the clinical settings [38]. In our present study, we therefore aimed to investigate the expression of key necroptotic proteins in a panel of CCA cell lines and evaluate necroptosis signaling as a novel potential therapeutic approach for CCA patients. More specifically, TNF-α signaling and a more relevant clinically used chemotherapeutic agent, gemcitabine, were evaluated in CCA cells. In addition, to develop a novel therapeutic strategy that enhances the therapeutic sensitivity of cells to gemcitabine, a Smac mimetic-based combinatory approach was also investigated in CCA cells. In this study, we demonstrated that RIPK1 and MLKL were expressed in a panel of CCA cell lines, in which the expression of MLKL in CCA cell lines was higher than that in nontumor cholangiocytes, while RIPK3 was differentially expressed in CCA cell lines. We further demonstrated that RIPK1/RIPK3/MLKL-dependent signaling is required for TNF-α and gemcitabine-induced necroptosis upon IAP depletion and caspase inhibition; therefore, our findings provide new insights toward designing a novel necroptosis-based therapeutic approach for CCA patients.

## Materials and method

### Cell line and culture

CCA cell lines (KKU213, KKU100, KKU214, KKU-M055, HuCCT-1) and MMNK1 were provided from the Japanese Collection of Research Bioresources (JCRB) Cell Bank, Osaka, Japan. HuCCA-1 [39] and RMCCA-1 [40] were developed from Thai patients with CCA. HT-29 and HEK293T were obtained from American Type Culture Collection (ATCC). All CCA cell lines

and MMNK1 were grown in HAM's F-12 medium (HyClone Laboratories, Logan, Utah, USA), while HT-29 and HEK293T were cultured in Dulbecco's modification of Eagle's medium (DMEM; HyClone Laboratories, Logan, Utah, USA) supplemented with 10% fetal bovine serum (Sigma, St Louis, Missouri, USA) and 1% penicillin streptomycin (HyClone Laboratories, Logan, Utah, USA) under the standard protocol at 37˚C in 5% $CO_2$ humidified atmosphere. All cultures were tested for mycoplasma contamination and were mycoplasma-free.

## Reagents and antibodies

Pan-caspase inhibitor, z-VAD-FMK (carbobenzoxy-valyl-alanyl-aspartyl-[O-methyl]- fluoro-methylketone), GSK'782, and necrosulfonamide (NSA) were purchased from Calbiochem (Merck Millipore, Darmstadt, Germany). Gemcitabine and Necrostatin-1 (Nec-1) were from Sigma (St Louis, Missouri, USA). TNF-α was from R&D systems (Minneapolis, Minnesota, USA). Smac mimetic (SM-164) was a gift from S. Wang (University of Michigan, Ann Arbor, Michigan, USA). Antibodies for Western blot were obtained from commercial sources: anti-RIPK1 (610459) and anti-FADD (610400) were from BD Biosciences (San Jose, California, USA); anti-RIPK3 (8457), anti-cIAP1 (7065), anti-cIAP2 (3130), anti-NIK (4994), anti-cyclin D1 (2978), anti-caspase-8 (9746) and anti-actin (4970) were from Cell Signaling (Danvers, Massachusetts, USA); anti-MLKL (ab184718) and anti-phosphorylated MLKL (ab187091) were from Abcam (Cambridge, UK).

## Different expression of key necroptotic proteins between CCA tumor tissues and normal bile ducts

RNA sequencing data obtained from The Cancer Genome Atlas (TCGA) dataset was analyzed using an online tool called Gene Expression Profiling Interactive Analysis (GEPIA) (http://gepia.cancer-pku.cn/, updated by November 13, 2018) [41]. Different expression of RIPK1, RIPK3 and MLKL between CCA tumor tissues (n = 36) and normal bile ducts (n = 9) was analyzed and presented in a box plot. The cutoff of |Log2FC| was 1. The cut-off *p*-value was 0.01.

## CRISPR constructs, shRNAs and Lentivirus infection

The shRNA lentiviral plasmids were obtained from Sigma (St Louis, Missouri, USA). The shRNA against human MLKL (NM_152649.4) corresponds to the 3' untranslated region 2025–2045 (shMLKL1) and 1907–1927 (shMLKL2). CRISPR plasmids targeting human RIPK1 (NM_003804) and human RIPK3 (NM_006871) were generated according to Zhang's protocol [42]. The sequence for CRISPR-RIPK1 was 5'–CACCGGATGCACGTGCTGAAAGC CG–3' and CRISPR-RIPK3 was 5'–CAGTGTTCCGGGCGCAACAT–3'. All of the plasmid constructs were confirmed by DNA sequencing. To generate lentiviral particles, HEK293T were co-transfected with packaging plasmid (pCMV-VSV-G) and envelope plasmid (pCMV-dr8.2-dvpr) and either shRNA-non-targeting (shNT; pLKO.1puro) or shRNA-MLKL (shMLKL) or CRISPR-V2 or CRISPR-RIPK1 or CRISPR-RIPK3 plasmids. After 24 h, supernatants were collected and supernatants containing viral particles were filtered through a 0.45 μM sterile filter membrane (Merck Millipore, Darmstadt, Germany). The lentiviral preparation was then used to infect the cells with 8 μg/mL of polybrene (Merck Millipore, Darmstadt, Germany). After 24 h of infection, cells were selected with puromycin (Merck Millipore, Darmstadt, Germany) for a further 48 h.

## Western blot analysis

Cell were washed twice with ice-cold phosphate buffered saline (PBS; HyClone Laboratories, Logan, Utah, USA) and were lysed in RIPA buffer (Merck Millipore, Darmstadt, Germany) containing a proteinase inhibitor cocktail (Roche, Mannheim, Germany) on ice for 30 min. Total protein concentrations were determined by Bradford assay (Bio-Rad, Hercules, California, USA). Total proteins (20–50 µg) were separated by 10–20% SDS-PAGE and proteins were transferred onto PVDF membranes. The membranes were blocked in 5% blotting-grade Blocker (Bio-Rad, Hercules, California, USA) at room temperature for 1 h, and the membranes were incubated with the primary antibodies at 4°C overnight. After incubation with primary antibodies, blots were washed three times with TBS-T (Tris-buffered saline, 0.5% Tween 20) buffer and incubated with horseradish peroxidase-conjugated secondary antibodies (Cell Signaling Technology, Danvers, Massachusetts, USA) at room temperature for 1 h. The proteins were visualized by enhanced chemiluminescence according to the manufacturer's instructions (Bio-Rad, Hercules, California, USA). All Western blots shown were representative of at least three independent experiments.

## Treatment, determination of cell death, and cell cycle analysis

Necroptosis was induced by TNF-α (10 ng/ml), Smac mimetic, SM-164 (10 nM) and zVAD-fmk (20 µM). Apoptosis was induced by TNF-α (10 ng/ml) and Smac mimetic (10 nM). In HuCCT-1 and KKU100, Smac mimetic was used at 25 nM. For gemcitabine experiments, cells were pretreated with Smac mimetic (5 nM) and zVAD-fmk (20 µM) for 2 h, after that the cells were treated with gemcitabine at different concentrations from 0.01, 0.1, 1, and 10 µM for 48 h (RMCCA-1) and 72 h (KKU213 and MMNK1). Cell death was determined by Annexin V-FITC and propidium iodide (PI) staining followed by flow cytometry. Briefly, cells were washed and resuspended in Annexin V binding buffer containing recombinant Annexin V-FITC (ImmunoTools, Friesoythe, Germany) and PI (Invitrogen, Carlsbad, California, USA). The stained cells were analyzed with flow cytometry (Navios, Beckman Coulter, Indianapolis, USA). Ten thousand events were collected for each sample and analyzed using Navios software. Combination index (CI) was calculated based on Chou-Talalay where CI = 1, CI < 1, and C > 1 indicates additive effect, synergism, and antagonism, respectively [43]. Cell cycle analysis was determined by propidium iodide (PI) staining of DNA content. Briefly, cells were fixed with 70% ethanol, after washing the cells were resuspended in PBS with 0.25% Triton X, containing RNase A (100 µg/ml) and PI (50 µg/ml) for 30 min. The stained cells were then analyzed by flow cytometry.

## RNA preparation, reverse transcription and real-time PCR

Total RNA was isolated from cells using TRIzol reagent (Invitrogen, Carlsbad, California, USA). A total of 1 µg RNA was reverse transcribed using RevertAid Reverse Transcriptase (Thermo fisher scientific, Waltham, Massachusetts, USA) with oligo (dT) 18 primer according to the manufacturer's protocol. Quantitative real-time PCR for TNF-α and GAPDH was performed by using *iTaq*™ universal *SYBR*® *Green* supermix (Bio-Rad, Hercules, California, USA) following the manufacturer's instructions. The following primers were used: TNF-α (Forward: 5′–GCCCATGTTGTAGCAAACCC–3′, Reverse: 5′–CTGATGGTGTGGGTGAGG AG–3′) and GAPDH (Forward: 5′–ACATCGCTCAGACACCATGG–3′, Reverse: 5′–ACCA GAGTTAAAA GCAGCCCT–3′). PCR cycling parameters were 95°C for 30 s, followed by 40 cycles of 95°C for 15 s and 57°C for 30 s and fluorescent signals were measured in real time. GAPDH was used as an internal control to normalize the amount of total RNA added to each

reaction and relative gene expressions were presented with the $2^{-\Delta\Delta Ct}$ method [44]. The results were expressed as fold induction over control cells.

## Statistical analysis

All statistical analyses were conducted using the software package SPSS for Windows. Results were expressed as the mean ± standard deviation (S.D.) of at least three independent experiments. Comparisons between two groups were determined by Student's t-test. Difference between two groups were considered statistically significant at $p$-value < 0.05 (*), $p$-value < 0.01 (**), and $p$-value < 0.001 (***).

## Results

### RIPK3, but not RIPK1 and MLKL, is differentially expressed in CCA cells

Loss of key necroptotic proteins has been shown to be a major hindrance for necroptosis-based treatments [22, 24, 29]; therefore, to evaluate the therapeutic potential of a novel necroptosis-based approach for CCA patients, we initially analyzed the mRNA expression of key necroptotic proteins, including RIPK1, RIPK3 and MLKL, in The Cancer Genome Atlas (TCGA) dataset using an online tool called GEPIA [41]. Both RIPK3 and MLKL mRNA were upregulated in CCA primary tissues compared to levels in normal bile ducts, while RIPK1 mRNA was similarly expressed between CCA primary tissues and normal bile ducts (Fig 1). We then determined the expression levels of key components of necroptotic proteins in CCA cell lines as an *in vitro* model study. We analyzed and compared the expression of RIPK1, RIPK3 and MLKL in a panel of human CCA cell lines and in immortalized nontumor cholangiocytes by Western blot analysis. All tested CCA cell lines and the nontumor cholangiocytes exhibited a similar expression level of RIPK1, while MLKL was slightly expressed in the nontumor cholangiocytes and was expressed at a higher level in all tested CCA cell lines (Fig 2A). There was no significant difference in the expression of RIPK1 and MLKL across different CCA cell lines. In marked contrast, RIPK3 was more highly expressed in 2/7 CCA cell lines (HuCCT-1 and RMCCA-1), lowly expressed in 2/7 CCA cell lines (KKU213 and HuCCA-1), and undetectable in 3/7 CCA cell lines (KKU100, KKU214, and KKU-M055) and the nontumor cholangiocytes (MMNK-1). In addition, two other proteins present in the death complex, Fas-associated protein with death domain (FADD) and caspase-8, were expressed at similar levels in all tested CCA cell lines and the nontumor cholangiocytes (Fig 2B). Our results demonstrated that RIPK3, but not RIPK1 and MLKL, was differentially expressed in CCA cells.

### CCA cells are sensitive to TNF-α/Smac mimetic-induced cell death upon caspase inhibition

To investigate whether CCA cells are sensitive to necroptosis treatment, we initially made use of a well-studied model of necroptosis, which is TNF-α signaling [9, 13]. Because RIPK3 expression is differentially expressed in CCA cell lines and it has been previously demonstrated that RIPK3 expression predicts necroptosis response in several cancers [13, 29], we hypothesized that the expression of RIPK3 determines necroptosis responsiveness in CCA cells. To this end, we selected RIPK3-expressing KKU213, RMCCA-1 and HuCCT-1 cell lines and RIPK3-nonexpressing KKU100 and KKU214 cell lines for analysis of necroptosis responsiveness. These five CCA cell lines exhibited similar levels of RIPK1, MLKL, FADD, and caspase-8 expression (Fig 2A and 2B). An immortalized cholangiocyte cell line, MMNK-1, was included as a nontumor cholangiocyte control. TNF-α-induced cell death or survival depends on the cellular context, here, we demonstrated that treatment with TNF-α alone for 24 h and 48 h did

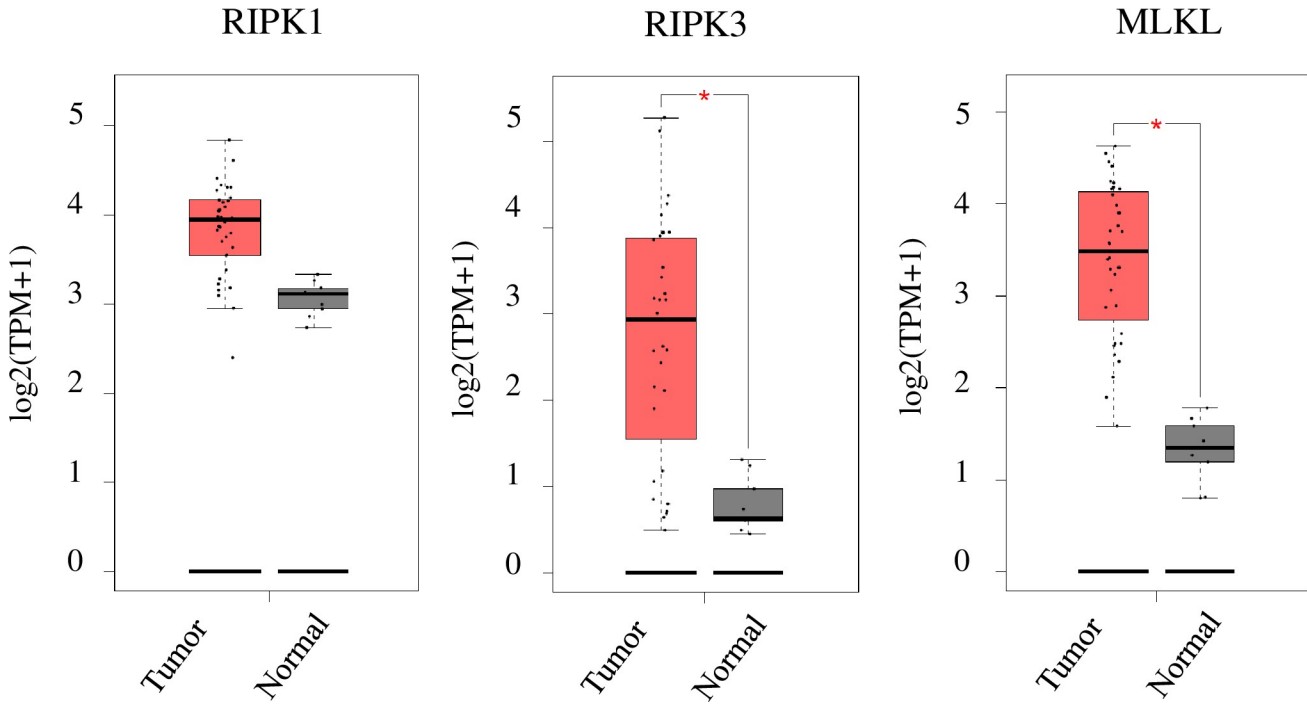

**Fig 1. Different expression of key necroptotic proteins including RIPK1, RIPK3 and MLKL in CCA tissues.** The mRNA expression of RIPK1, RIPK3, and MLKL was obtained from 36 CCA patients and 9 normal bile ducts. RNA sequencing data were retrieved from The Caner Genome Atlas (TCGA) and were analyzed by an online tool GEPIA. The red boxplot indicates tumor and the grey boxplot indicates normal tissue. The cutoff of |Log2FC| was 1 and the cut-off * indicates *p*-value < 0.01.

not induce cell death, while treatment with TNF-α and the Smac mimetic SM-164 (TS) dramatically induced cell death in all cell lines tested, although with a lower response in HuCCT-1 (Fig 2C and 2D). Interestingly, the addition of zVAD-fmk completely inhibited TNF-α and Smac mimetic-induced cell death in cell lines lacking RIPK3 expression, including MMNK-1, KKU100, and KKU214 cells, while CCA cell lines expressing RIPK3, including KKU213, RMCCA-1, and HuCCT-1 cells, were still sensitive to TNF-α/Smac mimetic/zVAD-fmk (TSZ)-induced cell death. Optimal concentrations of both TNF-α and the Smac mimetic were selected based on IC50 analysis at 24 h (S1 Fig). We observed under the microscope that treatment with TNF-α/Smac mimetic/zVAD-fmk induced cellular morphological changes resembling necrosis, in which the cells swelled and rounded-up; there was disruption of plasma membrane integrity in RIPK3-expressing cells, but no morphological changes were observed in RIPK3-nonexpressing cells (S2 Fig). Similar results in terms of cell death were detected with BV6, a bivalent Smac mimetic (S3 Fig). Unexpectedly, although HuCCT-1 cells exhibited the strongest expression of RIPK3, their responsiveness to TNF-α/Smac mimetic/zVAD-fmk-induced cell death with increased concentrations of Smac mimetic that completely abrogated cIAP1/2 expression (S4 Fig) was still lower than that in KKU213 and RMCCA-1 cells (Fig 2C and 2D, S1 Fig). We therefore hypothesized that there might be negative regulators that suppress both TNF-α/Smac mimetic-induced cell death and TNF-α/Smac mimetic/zVAD-fmk-induced cell death in the HuCCT-1 cell line. Cellular FLIP$_L$ (cFLIP$_L$) has been reported to inhibit the association of FADD, RIPK1, and caspase-8, a ripoptosome assembly that can mediate either apoptosis or necroptosis [45]. We demonstrated that cFLIP$_L$ exhibited the highest expression in HuCCT-1 cells when compared to other cells. Therefore, high expression of cFLIP$_L$ in HuCCT-1 cells might contribute to the marginal cell death induction (S5 Fig), but

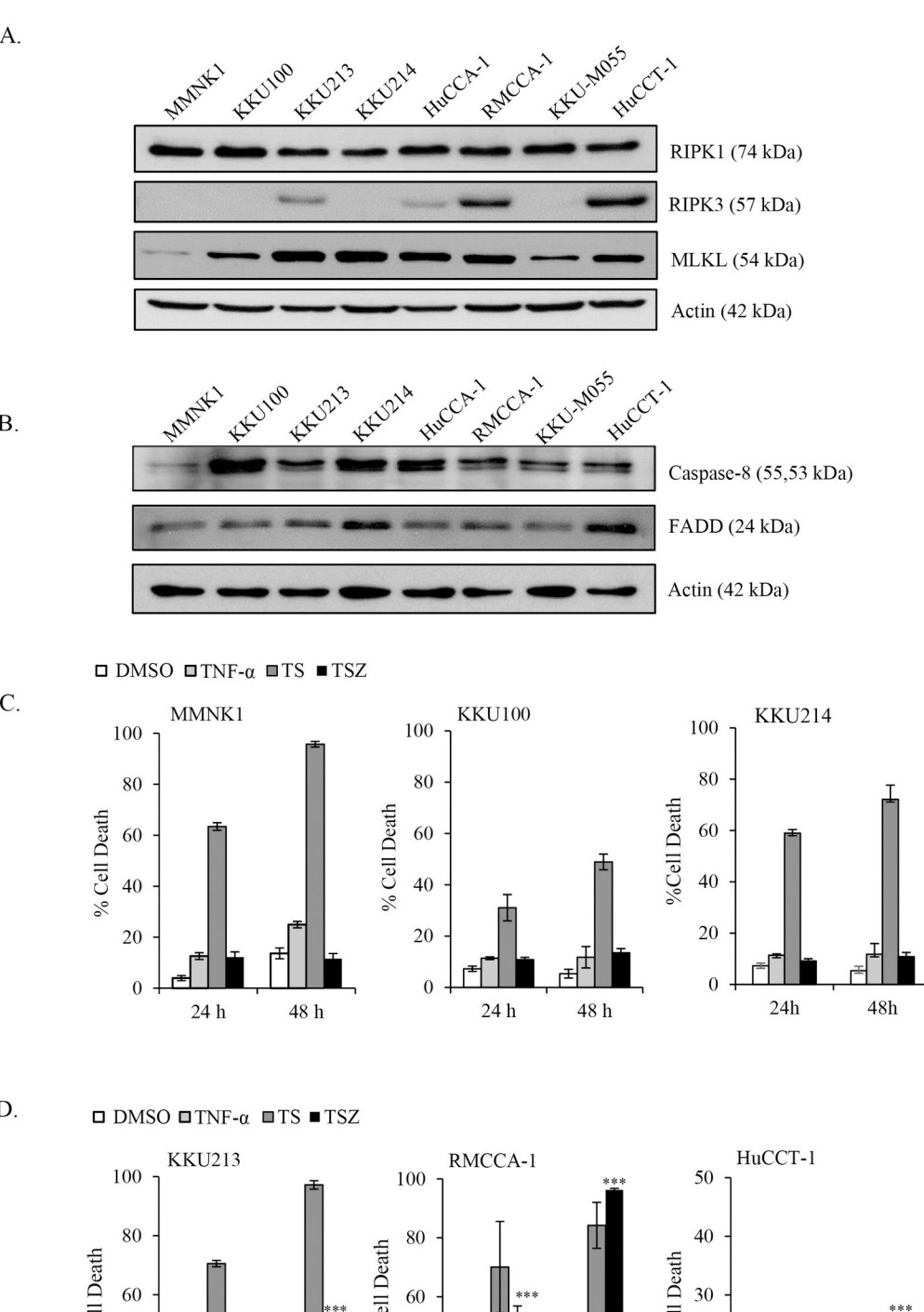

**Fig 2. RIPK1, RIPK3 and MLKL-expressing CCA cell lines are sensitive to TNF-α/Smac mimetic-induced cell death upon caspase inhibition.** (A) Key necroptotic proteins; RIPK1, RIPK3 and MLKL (B) Protein expression of FADD and caspase-8 were analyzed in 7 different human CCA cell lines and a nontumor human cholangiocyte cell line, MMNK1

using Western blot analysis and β-actin served as loading control. (C) RIPK3 deficient cells, MMNK1, KKU100, and KKU214 were treated with 10 ng/ml TNF-α (TNF-α, complex I), TNF-α and Smac mimetic, SM-164 (10 nM in MMNK1 and KKU214, or 25 nM in KKU100) (TS, complex IIa, apoptosis), or TNF-α and Smac mimetic in the presence of 20 μM zVAD-fmk (TSZ, complex IIb, necroptosis) for 24 h and 48 h. (D) RIPK3-expressing cells were treated as in C except KKU213 and RMCCA-1 were treated with 10 nM Smac mimetic and HuCCT-1 were treated with 25 nM Smac mimetic. Smac mimetic and zVAD-fmk were pretreated for 2 h followed by treatment with TNF-α for 24 h and 48 h. Percentage of cell death (AnnexinV+/PI- and AnnexinV+/PI+) were determined by Annexin V and propidium iodide (PI) staining and flow cytometry. Data presented as mean ± S.D. of three independent experiments are shown; * $p < 0.05$, ** $p < 0.01$, *** $p < 0.001$.

this idea needs further investigation, such as a genetic disruption of cFLIP$_L$ expression. Together, our results demonstrated that TNF-α/Smac mimetic/zVAD-fmk specifically induced cell death in RIPK3-expressing CCA cell lines, but this cell death was restricted to RIPK3-deficient cell lines, suggesting that the induction of cell death correlates with RIPK3 expression.

## Key necroptotic proteins are required for TNF-α/Smac mimetic-induced necroptosis upon caspase inhibition

To investigate whether TNF-α/Smac mimetic/zVAD-fmk trigger necroptosis, we used different approaches to validate necroptosis induction. Since phosphorylation of MLKL by RIPK3 has been suggested as a critical step for necroptosis execution [16, 17], we examined the phosphorylation of MLKL in MMNK-1, KKU213 and RMCCA-1 cell lines upon TNF-α/Smac mimetic/zVAD-fmk treatment at 8 h and 16 h. Time course analysis of MLKL phosphorylation was performed at 8 h and 16 h based on the observation of swelling and rounding-up of the cells that started at approximately 8 h, indicating the formation of the necrosome complex and the presence of phosphorylated MLKL. Phosphorylation of MLKL (phosphor-S358) was detected in KKU213 and RMCCA-1 cells after 8 h and 16 h but not in MMNK-1 cells (Fig 3A). The phosphorylation of MLKL was lower at 16 h when compared to the level at 8 h, probably due to the translocation of phosphorylated MLKL into the RIPA-insoluble pellet fraction. The higher RIPK3 expression and cell death in RMCCA-1 compared to KKU213 cells correlated with increased phosphorylation of MLKL. To further confirm necroptosis induction, we asked whether TNF-α/Smac mimetic/zVAD-fmk-induced cell death depended on key necroptotic proteins. To this end, pharmacological inhibitors of necroptotic proteins as well as genetic approaches were used to inhibit necroptotic protein function. RIPK1 (necrostatin-1, Nec-1), RIPK3 (GSK'872) and MLKL (necrosulfonamide, NSA) inhibitors significantly reduced TNF-α/Smac mimetic/zVAD-fmk-induced cell death (Fig 3B). The cell viability after treatment with GSK'872 at 10 μM alone was slightly decreased, indicating the toxicity of this inhibitor; however, there was no significant protection at lower concentrations of GSK'872 (S6 Fig). In addition to pharmacological inhibitors, CRISPR/cas9-mediated deletion of RIPK1 and RIPK3 and short hairpin RNA (shRNA) silencing of MLKL by two distinct shRNA sequences were used to create genetic knockout and knockdown models in KKU213 and RMCCA-1 cells. Notably, the expression of RIPK1 and RIPK3 was completely absent in both KKU213 and RMCCA-1 cells infected with CRISPR-RIPK1 and CRISPR-RIPK3 (Fig 3C). Similarly, knockdown of MLKL almost completely reduced MLKL levels (Fig 3D). Consistent with the results with pharmacological inhibitors, knockout of RIPK1 and RIPK3 or knockdown of MLKL rendered the cells resistant to TNF-α/Smac mimetic/zVAD-fmk-induced cell death (Fig 3C and 3D). Altogether, these findings demonstrated that TNF-α/Smac mimetic/zVAD-fmk induced necroptosis in CCA cells and that this necroptosis was dependent on RIPK1/RIPK3/MLKL necroptotic activity.

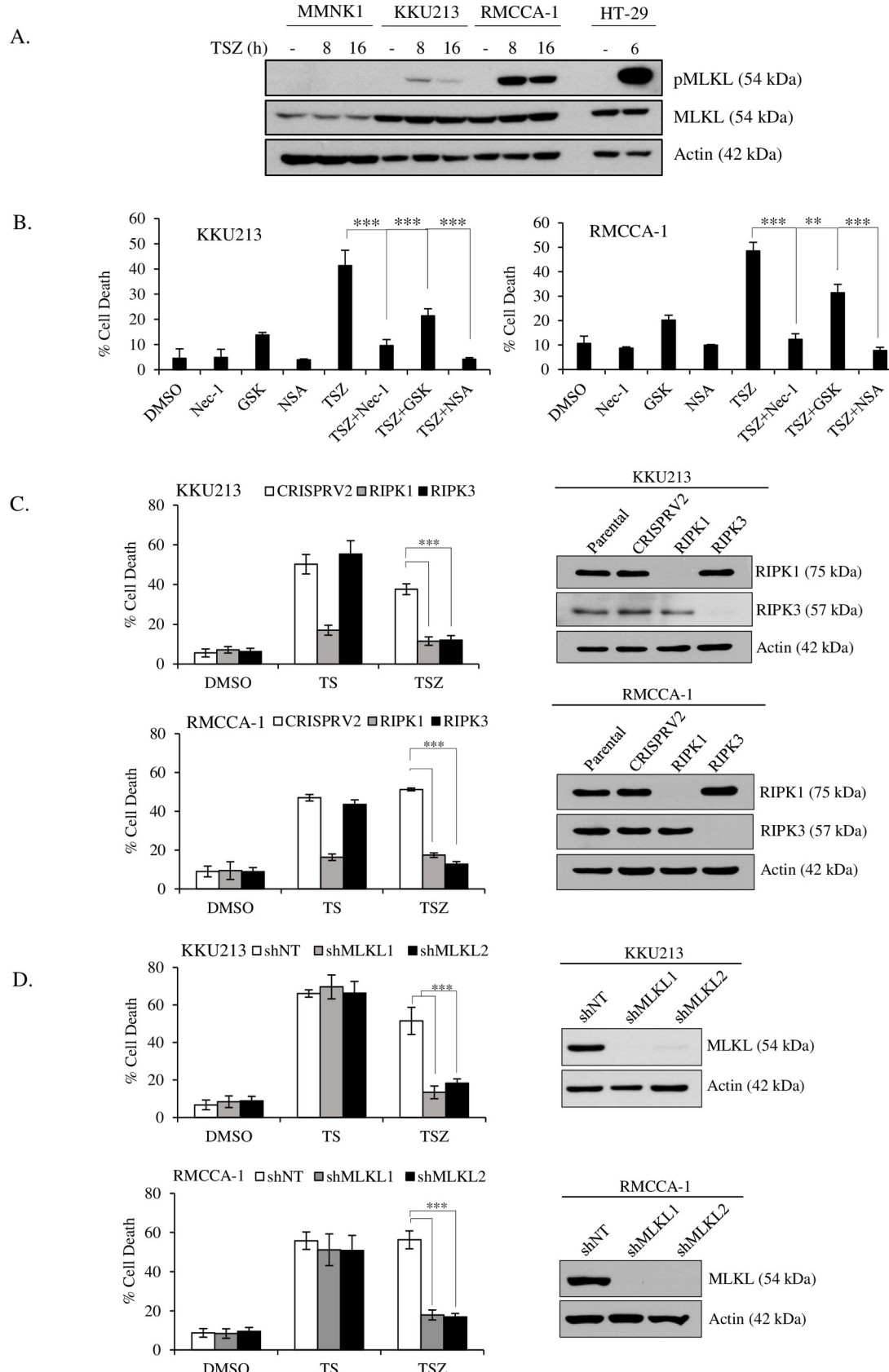

**Fig 3. TNF-α/Smac mimetic-induced cell death upon caspase inhibition is required RIPK1, RIPK3, and MLKL proteins.** (A) MMNK1, KKU213, and RMCCA-1 were treated with TSZ for 8 h and 16 h, phosphorylated MLKL (pMLKL) was analyzed by Western blot analysis, and β-actin and total MLKL served as loading control. HT-29 cells were included as a positive control for pMLKL (B) KKU213 and RMCCA-1 were pretreated with 60 μM necrostation-1 (Nec-1), 10 μM GSK'872 (GSK), or 1 μM necrosulfonamide (NSA) and Smac mimetic/zVAD-fmk for 2 h followed by TNF-α treatment for 24 h. (C) KKU213 and RMCCA-1 were infected with CRISPR-V2, CRISPR-RIPK1, or CRISPR-RIPK3 to generate RIPK1 and RIPK3 knockout cells. Cells were treated with TSZ for 24 h. The representative knockout efficiency was shown on right. (D) KKU213 and RMCCA-1 were infected with shRNA control (shNT) or shRNAs targeting two different sequences of MLKL (shMLKL1, shMLKL2). Cells were treated as in C. The representative knockdown efficiency was shown on right. Cell death was determined by Annexin V and PI staining and flow cytometry. Data presented as mean ± S.D. of three independent experiments are shown; * $p < 0.05$, ** $p < 0.01$, *** $p < 0.001$.

## Smac mimetic and gemcitabine synergistically trigger necroptosis upon caspase inhibition

Recent evidence suggests that standard chemotherapeutic drugs induce RIPK3-dependent necroptosis in multiple cancer cell lines [24] and that chemotherapeutic drug-induced necroptosis rather than apoptosis reduces tumor growth *in vivo* [46]. In contrast to previous studies, we found that gemcitabine induced caspase-independent cell death and was independent of the key necroptotic proteins RIPK1, RIPK3, and MLKL (S7 Fig). Accumulating evidence shows that small molecules that antagonize IAPs, Smac mimetics, greatly increase chemotherapy response to both apoptosis and necroptosis in cancer cells, although necroptosis has been far less examined [47–59]. We first examined the expression of cIAP1 and cIAP2 in a panel of CCA cells and found that both cIAP1 and cIAP2 were differentially expressed in CCA cell lines (S5 Fig). To search for a novel sensitizer that enhances gemcitabine treatment effectiveness and for to aid in the development of a novel necroptosis-based therapeutic approach with a potential clinical application for CCA, we hypothesized that a Smac mimetic could improve the sensitivity of CCA cells to gemcitabine-induced cell death. To prove our hypothesis, we treated KKU213, RMCCA-1, and MMNK-1 cells with different concentrations of gemcitabine (0, 0.01, 0.1, 1, and 10 μM) in the presence or absence of a low dose of a Smac mimetic, SM-164 (at 5 nM), when caspases were inhibited. We found that a low dose of the Smac mimetic minimally induced cell death in all tested cell lines (Fig 4A), while it greatly reduced cIAP1 and cIAP2 protein levels and stabilized NIK (S8 Fig). However, when the Smac mimetic was combined with gemcitabine in the presence of a caspase inhibitor, cell death was dramatically increased when compared to that seen with single-agent gemcitabine treatment in CCA cell lines, while the Smac mimetic alone did not sensitize MMNK-1 cells to gemcitabine-induced cell death (Fig 4A). Calculation of the combination index (CI) confirmed that the effect between the Smac mimetic and gemcitabine was highly synergistic (Fig 4A). Importantly, this combination treatment significantly enhanced cell death with a relatively low dose of gemcitabine. To further investigate whether the combination treatment triggered necroptosis in CCA cell lines, we used genetic approaches to knockout or knockdown the expression of key necroptotic proteins that had been used in previous experiments. Although key necroptotic proteins were dispensable for gemcitabine-induced cell death (S7 Fig), knockout of RIPK1 and RIPK3 and knockdown of MLKL reduced gemcitabine and Smac mimetic-induced cell death when caspases were inhibited to the level of cell death seen with single-agent gemcitabine treatment (Fig 4B and 4C). Furthermore, the combination treatment greatly increased MLKL phosphorylation in CCA cell lines (Fig 4D). Altogether, this set of experiments demonstrated that Smac mimetic sensitized CCA cells treated with a relatively low dose of gemcitabine to necroptosis when caspases were inhibited.

A.

B.

C.

D.

**Fig 4. Smac mimetic and gemcitabine synergistically trigger necroptosis upon caspase inhibition.** (A) MMNK1, KKU213, and RMCCA-1 were pretreated with DMSO or 5 nM Smac mimetic for 2 h in the presence of 20 μM zVAD-fmk followed by addition of 0.01, 0.1, 1, or 10 μM gemcitabine for 72 h (MMNK1 and KKU213) and 48 h (RMCCA-1). RIPK1 and RIPK3 knockout or MLKL knockdown (B) KKU213 and (C) RMCCA-1 cells were pretreated with DMSO or 5 nM Smac mimetic and 20 μM zVAD-fmk for 2 h followed by treatment with 1 μM gemcitabine for 72 h (KKU213) and 48 h (RMCCA-1). Cell death was determined by Annexin V and PI staining and flow cytometry. Data presented as mean ± S.D. of three independent experiments are shown; * $p < 0.05$, ** $p < 0.01$, *** $p < 0.001$. (D) KKU213 and RMCCA-1 were pretreated with DMSO or 5 nM Smac mimetic and 20 μM zVAD-fmk for 2 h followed by treatment with 1 μM gemcitabine as indicated times. Phosphorylated MLKL (pMLKL) was analyzed by Western blot analysis, and β-actin and total MLKL served as loading control.

## Smac mimetic and gemcitabine synergistically induce TNF-α mRNA levels

To unravel the molecular mechanisms underlying the synergistic effect of Smac mimetic and gemcitabine-induced necroptosis when caspases were inhibited, we focused our mechanistic studies on the following possible mechanisms. Smac mimetics have been shown to induce noncanonical NF-κB activation, leading to autocrine TNF-α production [33, 34]. Consistently, DNA-damaging agents induce cytokine production, including TNF-α, through canonical NF-κB activation [60]. Therefore, we hypothesized that a Smac mimetic and gemcitabine could enhance TNF-α production in CCA cells. To this end, IκBα degradation, NIK stabilization and TNF-α mRNA expression levels were analyzed after single drug or combination treatment in CCA cell lines. Gemcitabine treatment alone or the combination treatment for 24 h dramatically decreased IκBα in both KKU213 and RMCCA-1 cells, while the Smac mimetic alone or the combination treatment for 24 h greatly induced the accumulation of NIK in both KKU213 and RMCCA-1 cells (Fig 5A–5C). Single-agent treatment with either gemcitabine or the Smac mimetic slightly increased TNF-α mRNA production (Fig 5D). However, when gemcitabine was combined with the Smac mimetic in the presence of a caspase inhibitor, the production of TNF-α mRNA was further enhanced compared to that seen with single-agent treatment, and the result was more pronounced in RMCCA-1 cells (Fig 5D). These results indicated that the Smac mimetic and gemcitabine synergistically induced an increase in TNF-α mRNA levels.

## Smac mimetic reverses gemcitabine-induced S phase-arrested cells, leading to cell killing

Gemcitabine, an analog of deoxycytidine, incorporates into DNA, causing chain termination and inducing cell cycle arrest [61]. We therefore analyzed the cell cycle-perturbing effects of gemcitabine alone or in combination with a Smac mimetic. Gemcitabine markedly arrested CCA cells in S phase in both KKU213 (Fig 6A) and RMCCA-1 (Fig 6B) cells, while the Smac mimetic and a caspase inhibitor (zVAD-fmk) did not affect the cell cycle. Gemcitabine alone or in the combination treatment also decreased the expression of cyclin D1, a cell cycle regulator that is required for cell cycle progression through the G1 phase (Fig 6C). Interestingly, in the presence of the Smac mimetic and a caspase inhibitor, gemcitabine-induced S phase cell cycle arrest was reversed, and the number of cells in the sub-G1 phase was significantly increased. Therefore, our results indicated that CCA cells arrested in S phase following gemcitabine treatment probably died in the presence of the Smac mimetic and a caspase inhibitor, thereby correlating with the results of the cell death assay AnnexinV/PI staining (Fig 4A). These results suggested that a Smac mimetic and a caspase inhibitor can markedly enhance elimination of the S phase cell cycle arrest induced by gemcitabine.

## Discussion

CCA has low survival rates and poor prognosis due to limited therapeutic options. Here, we report for the first time that the key necroptotic proteins RIPK1, RIPK3 and MLKL, which have been reported to be lost or harbor reduced expression in other cancers, were expressed in both CCA primary tissues (TCGA database) and a panel of CCA cell lines. In line with these findings, CCA cell lines expressing key necroptotic proteins were sensitive to TNF-α/Smac mimetic/zVAD-fmk treatment. In this study, more than one CCA cell line was evaluated to confirm the generality of this particular finding, and of particular interest, induction of necroptosis was specific to CCA cell lines, but nontumor cholangiocytes were found to be resistant. Next, we demonstrated that this necroptosis induction was dependent on RIPK1, RIPK3, and MLKL based on experiments with both pharmacological inhibitors and genetic

A.

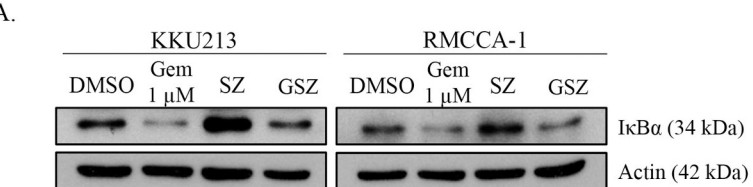

B.

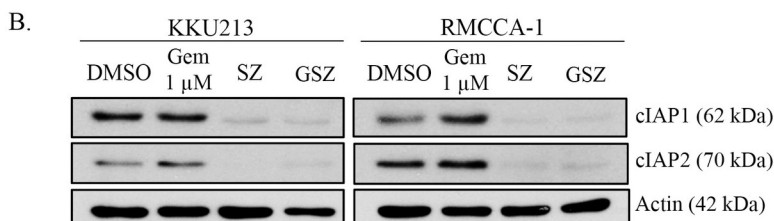

C.

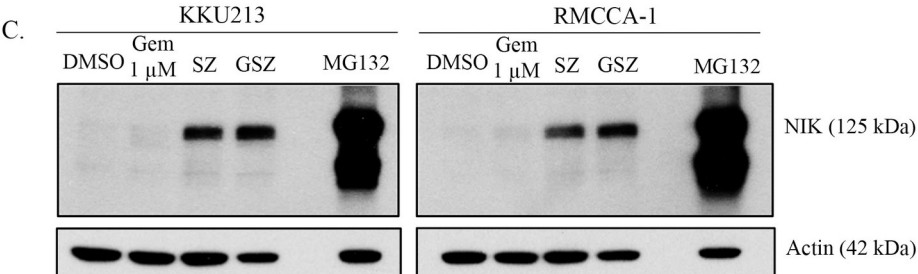

D.

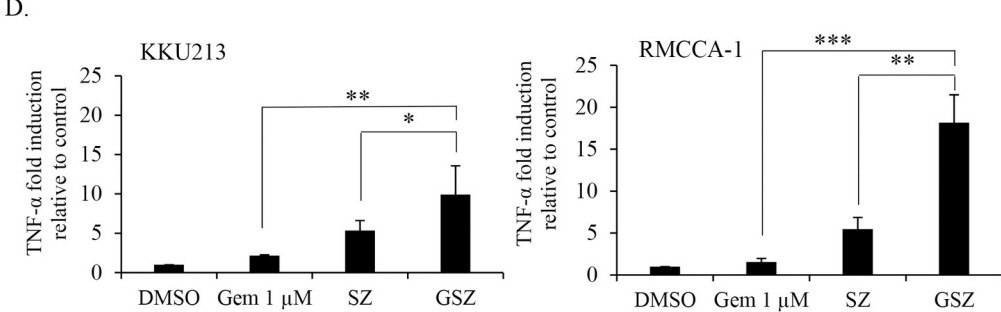

**Fig 5. Smac mimetic and gemcitabine synergistically induce TNF-α mRNA levels upon caspase inhibition.**
KKU213 and RMCCA-1 were treated with DMSO, 1 μM gemcitabine (Gem 1 μM), 5 nM Smac mimetic and 20 μM
zVAD-fmk (SZ), or GSZ for 24 h. The expression of IκBα (A), cIAP1, cIAP2 (B), and NIK (C) were determined by
Western blot analysis. MG132 (10 μM, 6 h) was used as a positive control for NIK stabilization. β-actin served as
loading control. (D) Cells were treated as in A, Total RNA was extracted and analyzed for TNF-α mRNA levels by
realtime PCR. Data were normalized to GAPDH levels and are expressed as fold induction of treated cells over
untreated cells (DMSO). Results shown are the average of triplicate measurement ± S.D. and obtained from two
independent experiments; $^*$ $p < 0.05$, $^{**}$ $p < 0.01$, $^{***}$ $p < 0.001$.

approaches. Most significantly, we showed that a Smac mimetic sensitized cells to necroptosis
in an RIPK1/RIPK3/MLKL-dependent manner at a relatively low dose of gemcitabine when
caspases were inhibited. The Smac mimetic and gemcitabine synergistically induced increases
in TNF-α mRNA levels, and the Smac mimetic reversed gemcitabine-induced cell cycle arrest,

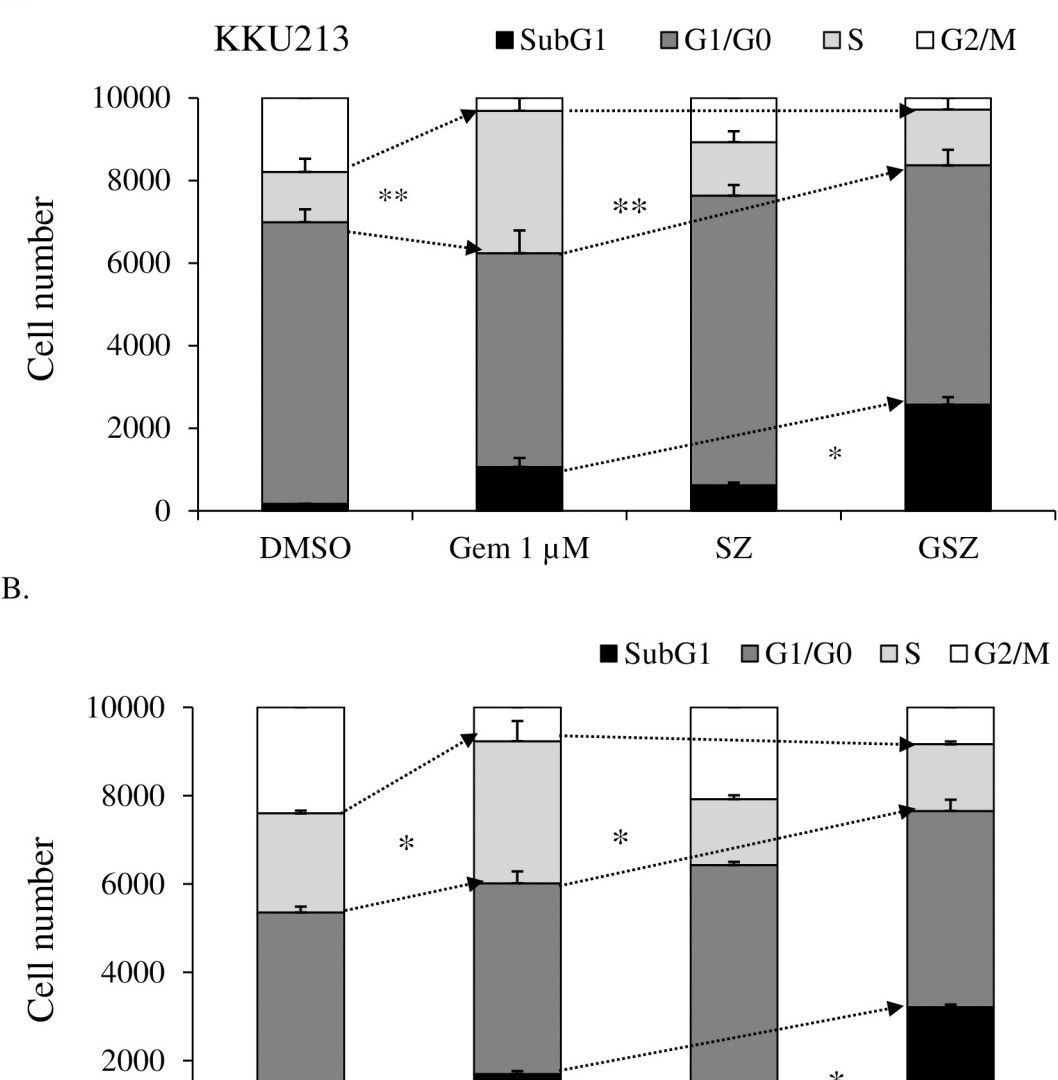

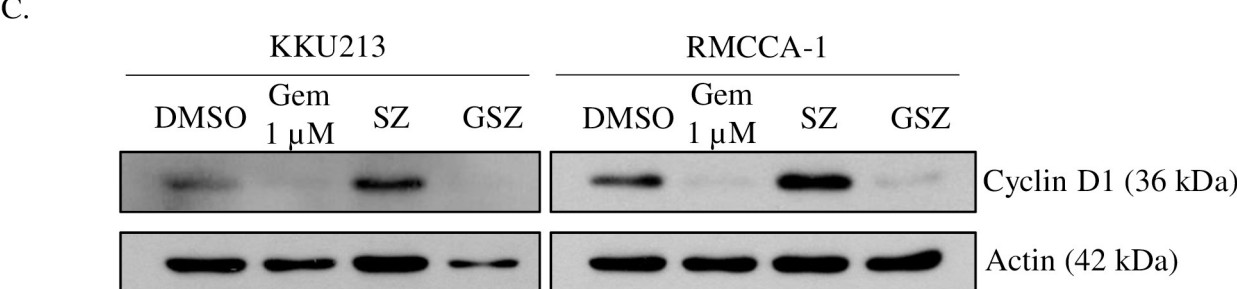

**Fig 6. Smac mimetic reverses gemcitabine-induced cell cycle arrest upon caspase inhibition.** (A) KKU213 and (B) RMCCA-1 were treated with DMSO, 1 μM gemcitabine (Gem 1 μM), 5 nM Smac mimetic and 20 μM zVAD-fmk (SZ), or GSZ for 48 h. Cells were stained with propidium iodide and cell cycle (sub-G1, G0/G1, S, and G2/M) was analyzed by flow cytometry. Number of cells in each phase, data presented as mean ± S.D.

of two independent experiments are shown; * $p < 0.05$, ** $p < 0.01$, *** $p < 0.001$. (C) KKU213 and RMCCA-1 were treated as in A and B for 48 h, the expression of cyclin D1 was determined by Western blot analysis. β-actin served as loading control.

leading to cell killing. Our findings provide a novel therapeutic concept for the development of necroptosis-based therapeutic approaches for CCA patients.

Little is known about necroptosis signaling and key necroptotic proteins in CCA and the therapeutic potential of their manipulation. Therefore, the results of our present study provide the first demonstration of the expression of key necroptotic proteins in both CCA clinical specimens (TCGA database) and a panel of CCA cell lines. Although the mRNA expression of both RIPK3 and MLKL was upregulated in CCA tissues (n = 36) compared to the level in normal bile ducts (n = 9), there was no correlation between RIPK3 or MLKL expression and overall survival (OS) or disease-free survival (DFS) as analyzed by the GEPIA online tool (S9 Fig); this discrepancy is partly explained by the limited number of CCA patients in TCGA database. The loss of key necroptotic proteins, particularly RIPK3, has been reported in multiple cancer cell lines and primary tissues [13, 22–28]. We found that RIPK3 expression was not detected in 3 of 7 CCA cell lines or in immortalized nontumor cholangiocytes. The remaining question is whether this finding was due to selective pressures during the tissue culture process, particularly at later passages, that were not seen in the early passages of cancer cell lines derived from patients. In line with the findings of several studies, RIPK3 expression is silent in a majority of cancer cell lines [13, 62]. Therefore, it is critical to further investigate the expression of key necroptotic proteins and the level of phosphorylated MLKL, a specific marker of necroptosis in CCA clinical specimens. The analysis of RIPK3 expression in CCA primary tissues by immunohistochemistry has been previously reported, but RIPK3 protein was expressed in most CCA tissues, with lower levels in tumor tissues than in the paired normal liver tissues [63]. This may be because the previously reported study used paired normal liver tissues instead of paired normal cholangiocytes. The expression of RIPK1, RIPK3, and MLKL was also reported to be upregulated in pancreatic adenocarcinoma (PDA) [64]. Via *in vivo* deletion of RIPK3, the researchers of the study in PDA further indicated that necroptosis was a driver of PDA oncogenesis and progression. Thus, our data suggest that the induction of necroptosis is a promising therapeutic target for CCA, but further investigation is required to clarify its role in cancer development and progression before necroptosis-based therapy can be used in clinical settings. Targeting necroptosis as a novel therapeutic approach to overcome therapeutic failure in cancers has become very important, as growing evidence suggests that targeting this novel cell death pathway, so-called immunogenic cell death (ICD), has the dual benefits of killing tumor cells and inducing antitumor immunity, as demonstrated in colon cancer and melanoma models [20, 65–67].

The more immediate impact of our study is the finding that a Smac mimetic led to the sensitization of CCA cells to gemcitabine-induced necroptosis when caspases were inhibited, leading to necroptotic death of the tumor cells. Although gemcitabine is used as a chemotherapeutic agent for the treatment of CCA, it is minimally effective due to resistance in CCA patients and leads to considerable side effects [3]. The mechanisms of gemcitabine resistance in CCA have not been well studied, although evasion of apoptosis is considered a key factor of chemotherapy resistance [6, 7]. Myeloid leukemia cell differentiation protein 1 (Mcl-1) upregulation mediated through IL-6 signaling in CCA cells could also be a contributing factor conferring gemcitabine resistance [68, 69]. Several studies have demonstrated that standard chemotherapeutic drugs, including gemcitabine, can trigger necroptosis [24, 64]. Indeed, we did show that RIPK1, RIPK3 and MLKL were all dispensable for gemcitabine-induced CCA cell death, as evaluated by genetic interruption of RIPK1, RIPK3 and MLKL expression. To

improve gemcitabine efficacy and decrease its side effects, here, we reported for the first time that a Smac mimetic enhanced gemcitabine-induced necroptosis in CCA cell lines. Smac mimetic-mediated sensitization to chemotherapeutic agent-induced apoptosis and necroptosis has been previously reported in several cancer cells and has been reported to suppress tumor growth in preclinical mouse models and early clinical trials of various cancers [47–59, 70]. The results of a study in colon cancer in which the combination of a Smac mimetic and standard chemotherapeutic drugs, when a caspase inhibitor was blocked, were independent of necroptosis [25]. The discrepancy could be explained by the difference in chemotherapeutic drugs and cell types used.

How a Smac mimetic and gemcitabine cooperate to enhance the sensitivity to cell death warrants further investigation to determine the precise molecular mechanism. TNF-α mRNA expression was transcriptionally increased in CCA cells treated with a reduced dose of the Smac mimetic when caspases were inhibited, but this level appeared insufficient to induce massive cell death, as was seen with the TNFα/Smac mimetic/zVAD treatment in which exogenous TNF-α was added. This result also indicated that the pro-death signaling generated by the Smac mimetic treatment alone could not necessarily overcome cell proliferation, as the Smac mimetic did not significantly inhibit the cell cycle, allowing the growth of surviving cells. Extensive DNA damage caused by DNA-damaging agents activates the ATM/ATR DNA damage signaling pathway, which subsequently leads to the activation of canonical NF-κB signaling and TNF-α production [60]. In this study, we revealed that gemcitabine induced IκBα degradation, a marker of canonical NF-κB activation. Gemcitabine minimally increased TNF-α mRNA expression when compared to treatment with the Smac mimetic. However, the combined treatment of the Smac mimetic and gemcitabine synergistically increased TNF-α mRNA levels and dramatically enhanced cell death compared to single-agent treatment. These results all indicate that the combination of a Smac mimetic with gemcitabine could lead to the accumulation of TNF-α production. Upon cIAP1 and cIAP2 degradation by the Smac mimetic and in the presence of RIPK1, RIPK3 and MLKL, the accumulation of TNF-α production may serve as a pro-death signal to trigger the induction of necroptosis in CCA cells when caspases are also inhibited. However, the involvement of autocrine/paracrine TNF-α needs to be further verified by either neutralization or suppression of TNF-α production in these cells. Supporting evidence from a clinical trial of a Smac mimetic, DEBIO1143, combined with daunorubicin and cytarabine in patients with acute myeloid leukemia (AML) revealed that the patients who responded more frequently showed an increase in plasma TNF-α levels [70].

Our data suggest that the sensitization to cell death induced by the combination treatment could also be explained by the ability of gemcitabine to inhibit cell proliferation and arrest cells in the S phase. As discussed above, because the pro-death signal has to be high enough to overcome cell proliferation, we therefore proposed that a low dose of gemcitabine could inhibit cell proliferation and cell cycle arrest, allowing the accumulation of pro-death signals received as a result of the combination treatment to orchestrate CCA cell death. Accordingly, the results of our present study represent a promising novel therapeutic approach for the improvement of gemcitabine-induced CCA cell death by Smac mimetics. Of particular importance, a low dose of gemcitabine was sufficient to trigger cell death due to the synergistic effects, which might decrease the side effects when gemcitabine is used in combination with a Smac mimetic. Several Smac mimetics currently have entered preclinical and early clinical trials [38]. Therefore, it is very likely that such a therapeutic concept could be translated into clinical applications. However, future *in vivo* studies are required to verify the therapeutic potential of necroptosis for gemcitabine and Smac mimetic-suppressed CCA tumor growth.

In conclusion, the results of our present study provide important implications for designing a novel necroptosis-based therapeutic approach for CCA patients. The necroptotic induction

by TNF-α signaling in a panel of CCA cells was dependent on RIPK1/RIPK3/MLKL. In addition, Smac mimetic sensitized cells to relatively low doses of gemcitabine-induced necroptosis in an RIPK1/RIPK3/MLKL-dependent manner. Because Smac mimetics such as LCL161 and DEBIO1143 are currently being investigated in clinical trials as monotherapy or in combination with chemotherapeutic drugs, our findings could lead to a novel potential therapeutic approach to improve the efficacy and decrease the side effects of gemcitabine for CCA patients.

## Supporting information

**S1 Fig. Time-course and dose-response analysis of TNF-α and Smac mimetic in the presence of zVAD-fmk treatment in RIPK3-expressing cell lines.** (A) KKU213 (B) RMCCA-1, and (C) HuCCT-1. Cells were pretreated with SZ (Smac mimetic, 1, 5, 10, 25, and 50 nM; zVAD-fmk, 20 μM) for 2 h, followed by treatment with T (TNF-α, 1, 10, 20 ng/ml) for 24 h and 48 h. Cell viability was determined by MTT assay. Inset indicates the concentration of TNF-α and Smac mimetic around IC50 at 24 h and was selected for further analysis.
(TIF)

**S2 Fig. Representative of cell morphology and representative of flow cytometry analysis.** (A) Representative of cell morphology upon treatment with TNF-α/Smac mimetic in the presence of zVAD-fmk in RIPK3-deficient cells (MMNK1, KKU100, and KKU214) and RIPK3-expressing cells (KKU213, RMCCA-1, and HuCCT-1). (B) Representative of flow cytometry analysis of cells, treated as in A.
(TIF)

**S3 Fig. RIPK1, RIPK3 and MLKL-expressing CCA cell lines are sensitive to TNF-α/ BV6-induced cell death upon caspase inhibition.** (A) KKU213 and (B) RMCCA-1 were treated with 10 ng/ml TNF-α, TNF-α and 5 μM BV6 (TB), or TNF-α and BV6 in the presence of 20 μM zVAD-fmk (TBZ) for 24 h and 48 h. Percentages of cell death (AnnexinV+/PI- and AnnexinV+/PI+) were determined by Annexin V and PI staining and flow cytometry. Data presented as mean ± S.D. of three independent experiments are shown; * $p < 0.05$, ** $p < 0.01$, *** $p < 0.001$
(TIF)

**S4 Fig. The sensitivity of HuCCT-1 to TNF-α-induced necroptosis.** HuCCT-1 cells were treated with different concentration of Smac mimetic (S) (0 nM, 25 nM, 50 nM, 100 nM) in the presence of 20 μM zVAD-fmk (Z) with or without 10 ng/ml TNF-α for 24 h and 48 h. Percentages of cell death (AnnexinV+/PI- and AnnexinV+/PI+) were determined by Annexin V and PI staining and flow cytometry. Data presented as mean ± S.D. of three independent experiments are shown; * $p < 0.05$, ** $p < 0.01$, *** $p < 0.001$
(TIF)

**S5 Fig. The expression of cFLIPL, cIAP1 and cIAP2.** (A) Seven CCA cells and a nontumor cholangiocyte, MMNK1 cell lysates were collected and subjected to Western blot analysis. β-actin was served as loading control. (B) cFLIP$_L$ was normalized to actin protein expression, and presented as fold increase relative to MMNK1 with its mean set to 1.
(TIF)

**S6 Fig. Dose responses of GSK'872 in the protection of TNF-α/Smac mimetic/zVAD-fmk-induced necroptosis.** (A) KKU213 and (B) RMCCA-1 were pretreated with 1 μM, 5 μM, and 10 μM of GSK'872 and Smac mimetic/zVAD-fmk for 2 h followed by treatment with 10 ng/ml TNF-α for 24 h. Percentages of cell death were determined by Annexin V and PI staining and

flow cytometry. Data presented as mean ± S.D. of three independent experiments are shown; * $p < 0.05$, ** $p < 0.01$, *** $p < 0.001$
(TIF)

**S7 Fig. Key necroptotic proteins are dispensable for gemcitabine-induced cell death.** (A) KKU213 and RMCCA1 were treated with 0.01, 0.1, 1, or 10 µM gemcitabine in the presence or absence of 20 µM zVAD-fmk for 72 h (KKU213) and 48 h (RMCCA-1). RIPK1 and RIPK3 knockout or MLKL knockdown (B) KKU213 and (C) RMCCA-1 cells were treated with 1 µM or 10 µM gemcitabine in the presence or absence of 20 µM zVAD-fmk for 72 h (KKU213) and 48 h (RMCCA-1). Cell death was determined by Annexin V and PI staining and flow cytometry. Percentages of cell death presented as mean ± S.D. of three independent experiments are shown.
(TIF)

**S8 Fig. Smac mimetic, SM-164 induces degradation of cIAP1 and cIAP2 and stabilization of NIK.** KKU213 and RMCCA-1 were treated with 5 nM Smac mimetic for indicated time points. The expression of cIAP1 and cIAP2 (A), and NIK (B) were determined by Western blot analysis. MG132 (10 µM, 6 h) was used as a positive control for NIK stabilization. β-actin served as loading control.
(TIF)

**S9 Fig. Kaplan-Meier analysis of the relationship between overall survival or disease free survival and RIPK3 or MLKL.** The association between overall survival or disease free survival and RIPK3 (A) or MLKL (B) expression was analyzed from GEPIA database. Samples with expression level higher than the median of TPM (transcripts of per million) are considered as the high-expression cohort (High). Samples with expression level lower than the median of TPM are considered the low-expression cohort (Low).
(TIF)

**S1 Raw Images. Raw images for all blots used in figures.** Full unedited images for Figs 2A, 2B, 3A, 3C, 3D, 4D, 5A–5C, 6C.
(PDF)

## Acknowledgments

We thank Dr. Zheng-Gang Liu (National Institutes of Health, Maryland, USA) for providing lentiviral expression system, CRISPR backbone construct, useful antibodies and reagents. We thank Prof. Satitaya Sirisinha (Mahidol University, Bangkok, Thailand); Assist. Prof. Dr. Panthip Rattanasinganchan (Huachiew Chalermprakiet University, Samut Prakan Province, Thailand); and Asst. Prof. Dr. Chanchai Boonla (Chulalongkorn University, Bangkok, Thailand) for kindly provided HuCCA-1; HuCCT-1; and KKU214, respectively.

## Author Contributions

**Conceptualization:** Siriporn Jitkaew.

**Data curation:** Perawatt Akara-amornthum, Thanpisit Lomphithak, Siriporn Jitkaew.

**Formal analysis:** Siriporn Jitkaew.

**Funding acquisition:** Perawatt Akara-amornthum, Rutaiwan Tohtong, Siriporn Jitkaew.

**Investigation:** Perawatt Akara-amornthum, Thanpisit Lomphithak, Siriporn Jitkaew.

**Methodology:** Perawatt Akara-amornthum, Thanpisit Lomphithak, Siriporn Jitkaew.

**Project administration:** Siriporn Jitkaew.

**Resources:** Swati Choksi, Rutaiwan Tohtong, Siriporn Jitkaew.

**Supervision:** Swati Choksi, Rutaiwan Tohtong, Siriporn Jitkaew.

**Validation:** Siriporn Jitkaew.

**Writing – original draft:** Siriporn Jitkaew.

**Writing – review & editing:** Swati Choksi, Rutaiwan Tohtong, Siriporn Jitkaew.

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
