## [Decision Letter · Decision Letter 0]

11 Sep 2019

PONE-D-19-20484

Key necroptotic proteins are required for Smac mimetic-mediated sensitization of cholangiocarcinoma cells to TNF-α and chemotherapeutic gemcitabine-induced necroptosis

PLOS ONE

Dear Dr. Jitkaew,

Thank you for submitting your manuscript to PLOS ONE. After careful consideration, we feel that it has merit but does not fully meet PLOS ONE’s publication criteria as it currently stands. Therefore, we invite you to submit a revised version of the manuscript that addresses the points raised during the review process.

Please address the comments of both Reviewers.

Additionally, the manuscript needs English language editing work.

We would appreciate receiving your revised manuscript by Oct 26 2019 11:59PM. To enhance the reproducibility of your results, we recommend that if applicable you deposit your laboratory protocols in protocols.io, where a protocol can be assigned its own identifier (DOI) such that it can be cited independently in the future. For instructions see: http://journals.plos.org/plosone/s/submission-guidelines#loc-laboratory-protocols

We look forward to receiving your revised manuscript.

Kind regards,

Irina V. Lebedeva, Ph.D.

Academic Editor

PLOS ONE

Journal Requirements:

Reviewers' comments:

Reviewer's Responses to Questions

**Comments to the Author**

1. Is the manuscript technically sound, and do the data support the conclusions?

Reviewer #1: Yes

Reviewer #2: Yes

2. Has the statistical analysis been performed appropriately and rigorously? 

Reviewer #1: Yes

Reviewer #2: Yes

3. Have the authors made all data underlying the findings in their manuscript fully available?

Reviewer #1: Yes

Reviewer #2: No

4. Is the manuscript presented in an intelligible fashion and written in standard English?

Reviewer #1: Yes

Reviewer #2: Yes

5. Review Comments to the Author

Reviewer #1: The paper by Akara-amornthum et al. provides strong, convincing evidence that cholangiocarcinoma cells can be killed by smac-mimetic plus gemcitabine if RIPK3 and MLKL are both present. The experiments are well performed, clearly described, and give strong support to the conclusions reached.

I only have some relatively minor criticisms/suggestions:

1. In figure 6 the y axis scale should stop at 100%. The authors should also discuss whether the apparent reversal of gemcytabine’s effect on the cell cycle are instead due to selective cell death. It would be interesting to see the absolute cell numbers, rather than the percentages.

2. I suggest the authors cite some more papers from other groups showing that smac mimetic induced IAP depletion, plus caspase inhibition, combined with chemotherapeutic agents such as gemcitabine, cause necroptosis in other cancer cell lines, e.g.:

Requirement of nuclear factor κB for Smac mimetic-mediated sensitization of pancreatic carcinoma cells for gemcitabine-induced apoptosis.

Stadel D, Cristofanon S, Abhari BA, Deshayes K, Zobel K, Vucic D, Debatin KM, Fulda S.

Neoplasia. 2011 Dec;13(12):1162-70.

Smac mimetic and demethylating agents synergistically trigger cell death in acute myeloid leukemia cells and overcome apoptosis resistance by inducing necroptosis.

Steinhart L, Belz K, Fulda S.

Cell Death Dis. 2013 Sep 12;4:e802. doi: 10.1038/cddis.2013.320.

Brumatti G, Ma C, Lalaoui N, Nguyen NY, Navarro M, Tanzer MC, Richmond J, Ghisi M, Salmon JM, Silke N, Pomilio G, Glaser SP, de Valle E, Gugasyan R, Gurthridge MA, Condon SM, Johnstone RW, Lock R, Salvesen G, Wei A, Vaux DL, Ekert PG, Silke J.

Sci Transl Med. 2016 May 18;8(339):339ra69. doi: 10.1126/scitranslmed.aad3099.

Birinapant (TL32711) Improves Responses to GEM/AZD7762 Combination Therapy in Triple-negative Breast Cancer Cell Lines.

Min DJ, He S, Green JE.

Anticancer Res. 2016 Jun;36(6):2649-57.

Reviewer #2: Overview

This manuscript presents data demonstrating that necroptosis can be induced to kill CCA cell lines providing they have not silenced the expression of RIPK3, and that concurrent addition of Smac mimetic can produce additional cell death to that induced by conventional CCA chemotherapeutic gemcitabine. Gemcitibine amplifies the capacity for TNFa gene expression beyond an additive effect in one cell line tested.

General comments/suggestions/questions;

This manuscript is very nicely written but will require some additional editing for English grammar in parts.

These results are original and do not appear to have been published elsewhere.

The experiments have been performed to a very high standard and are very clearly presented.

Please provide all independent data points for all figured with error bars (in accordance with PLoS one policy)

From the cell data presented in Figure 4A, it is really not convincing that the smac mimetic and gemcitabine work ‘synergisticly’. The only convincing synergy between thes two drugs is in their capacity to induce TNF-a mRNA production in RMCCA-1 cells in Fig. 5D. I recommend that all use of the word ‘synergy’ to describe 4A is replaced with ‘additive effect’.

I acknowledge that the authors clearly highlight that the analysis of patient-derived clinical samples will be necessary for confirmation of the findings presented here. Can the authors please include in their discussion section that that the loss of RIPK3 expression in long-held immortalized laboratory cell lines, and even newly derived MEF cell lines – is commonly observed in the cell death field (important references here are He et al, 2009 PMID:19524512 and Cook et all 2014 PMID:24902899), and that the absence of RIPK3 in CCA cell lines used may have occurred at a later stage as an artifact of cell culture and not in the patient samples they were derived from.

6. PLOS authors have the option to publish the peer review history of their article (what does this mean?). If published, this will include your full peer review and any attached files.

Reviewer #1: No

Reviewer #2: No

---

## [Author Response · Author response to Decision Letter 0]

25 Oct 2019

Our point-to-point responses to the reviewers are the following:

Reviewer #1 (Remarks to the Author):

The paper by Akara-amornthum et al. provides strong, convincing evidence that cholangiocarcinoma cells can be killed by smac-mimetic plus gemcitabine if RIPK3 and MLKL are both present. The experiments are well performed, clearly described, and give strong support to the conclusions reached.

I only have some relatively minor criticisms/suggestions:

1) In figure 6 the y axis scale should stop at 100%. The authors should also discuss whether the apparent reversal of gemcytabine’s effect on the cell cycle are instead due to selective cell death. It would be interesting to see the absolute cell numbers, rather than the percentages.

Response: In Figure 6, we have changed the cell cycle analysis data by Flow cytometry from the percentages to cell count and set maximum Y axis scale to 10,000 events (cells). 

In our manuscript (result section, the last part), we showed that gemcitabine induced S phase cell cycle arrest. However, in the presence of Smac mimetic and zVAD, we found a decrease in the S phase population, while a concomitant increase in a sub-G1 population indicating dead cells. These results are correlated with Annexin V/PI staining shown in Figure 4A. However, in the current manuscript we did not prove yet that gemcitabine-induced S phase cell cycle arrest, those S phase cells are actually killed by Smac mimetic and zVAD. We are currently working in this part and think this will be another story.

2) I suggest the authors cite some more papers from other groups showing that smac mimetic induced IAP depletion, plus caspase inhibition, combined with chemotherapeutic agents such as gemcitabine, cause necroptosis in other cancer cell lines

Response: We have cited more papers as a reviewer’s suggestion. More citations are added in result section, line 325 and in discussion section, line 449 (Ref 56, 57, 58, 59)

Reviewer #2 (Remarks to the Author):

This manuscript presents data demonstrating that necroptosis can be induced to kill CCA cell lines providing they have not silenced the expression of RIPK3, and that concurrent addition of Smac mimetic can produce additional cell death to that induced by conventional CCA chemotherapeutic gemcitabine. Gemcitibine amplifies the capacity for TNFa gene expression beyond an additive effect in one cell line tested.

General comments/suggestions/questions;

1) This manuscript is very nicely written but will require some additional editing for English grammar in parts.

Response: Our manuscript was edited for proper English language, grammar, punctuation, spelling, and overall style by one or more of the highly qualified native English speaking editors at AJE. The certificate was issued on October 9, 2019 and may be verified on the AJE website using the verification code 0762-4932-0380-8C06-34A9.

2) These results are original and do not appear to have been published elsewhere.

-

3) The experiments have been performed to a very high standard and are very clearly presented.

- 

4) Please provide all independent data points for all figured with error bars (in accordance with PLoS one policy)

Response: We have done experiments at least 3 independently repeated experiments

and statistically presented as mean (but not plotting individual data points) + error bars. However, error bars in some experiments are very small and cannot be seen clearly. Therefore, we have made some changes to the scale in order to see clear error bars in accordance with PLoS one policy.

5) From the cell data presented in Figure 4A, it is really not convincing that the smac mimetic and gemcitabine work ‘synergisticly’. The only convincing synergy between thes two drugs is in their capacity to induce TNF-a mRNA production in RMCCA-1 cells in Fig. 5D. I recommend that all use of the word ‘synergy’ to describe 4A is replaced with ‘additive effect’.

Response: We interpreted synergistic/additive interaction by using Combination index (CI) that was calculated based on Chou-Talalay where CI = 1, CI < 1, and C > 1 indicates additive effect, synergism, and antagonism, respectively (Chou T-C, 2010), see materials and methods. This program has been cited by many previous publications. To our knowledge, Synergistic interaction means that the effect of two chemicals taken together is greater than the sum of their separate effect at the same doses. Additive interaction means the effect of two chemicals is equal to the sum of the effect of the two chemicals taken separately. According to results in Figure 4A, for example in KKU213 cell line, the highest synergistic based on Combination index (CI) was Smac+zVAD (SZ) at 5 nM and gemcitabine at 1 µM in which CI index was 0.09. Percentage of cell death: Smac+zVAD (SZ) at 5 nM = 10.75%, gemcitabine at 1 µM = 35.51%, and the combination of Smac+zVAD (SZ) and gemcitabine = 66.5%. According to the definition of synergistic interaction, the % cell death from the combination 66.5% is greater than the sum of Smac+zVAD (SZ) alone and gemcitabine alone (10.75+35.51 = 46.26%).

6) I acknowledge that the authors clearly highlight that the analysis of patient-derived clinical samples will be necessary for confirmation of the findings presented here. Can the authors please include in their discussion section that that the loss of RIPK3 expression in long-held immortalized laboratory cell lines, and even newly derived MEF cell lines – is commonly observed in the cell death field (important references here are He et al, 2009 PMID:19524512 and Cook et all 2014 PMID:24902899), and that the absence of RIPK3 in CCA cell lines used may have occurred at a later stage as an artifact of cell culture and not in the patient samples they were derived from.

Response: We appreciated the reviewer for this important comment. We have added this part (possibility of loss of RIPK3 expression in late passage) in our discussion part (line 412-416) and cited two of publications as the reviewer’s suggestion.

---

## [Decision Letter · Decision Letter 1]

19 Dec 2019

Key necroptotic proteins are required for Smac mimetic-mediated sensitization of cholangiocarcinoma cells to TNF-α and chemotherapeutic gemcitabine-induced necroptosis

PONE-D-19-20484R1

Dear Dr. Jitkaew,

We are pleased to inform you that your manuscript has been judged scientifically suitable for publication and will be formally accepted for publication once it complies with all outstanding technical requirements.

With kind regards,

Irina V. Lebedeva, Ph.D.

Academic Editor

PLOS ONE

Additional Editor Comments (optional):

Reviewers' comments:

Reviewer's Responses to Questions

**Comments to the Author**

1. If the authors have adequately addressed your comments raised in a previous round of review and you feel that this manuscript is now acceptable for publication, you may indicate that here to bypass the “Comments to the Author” section, enter your conflict of interest statement in the “Confidential to Editor” section, and submit your "Accept" recommendation.

Reviewer #1: All comments have been addressed

2. Is the manuscript technically sound, and do the data support the conclusions?

Reviewer #1: Yes

3. Has the statistical analysis been performed appropriately and rigorously? 

Reviewer #1: Yes

4. Have the authors made all data underlying the findings in their manuscript fully available?

Reviewer #1: Yes

5. Is the manuscript presented in an intelligible fashion and written in standard English?

Reviewer #1: Yes

6. Review Comments to the Author

Reviewer #1: (No Response)

7. PLOS authors have the option to publish the peer review history of their article (what does this mean?). If published, this will include your full peer review and any attached files.

Reviewer #1: No

---

## [Editor Report · Acceptance letter]

23 Dec 2019

PONE-D-19-20484R1 

Key necroptotic proteins are required for Smac mimetic-mediated sensitization of cholangiocarcinoma cells to TNF-α and chemotherapeutic gemcitabine-induced necroptosis 

Dear Dr. Jitkaew:

I am pleased to inform you that your manuscript has been deemed suitable for publication in PLOS ONE. Congratulations! Your manuscript is now with our production department. 

With kind regards,

on behalf of

Dr. Irina V. Lebedeva 

Academic Editor

PLOS ONE